complexity

urban scaling, Twitter, linguistic laws

**Author for correspondence:**
Eszter Bokányi
e-mail: bokanyi@complex.elte.hu

# Scaling in words on Twitter

Eszter Bokányi[1], Dániel Kondor[2,3] and Gábor Vattay[1]

[1]Eötvös Loránd University, Budapest, Hungary
[2]Senseable City Laboratory, MIT, Cambridge, MA 02139, USA
[3]Singapore-MIT Alliance for Research and Technology, Singapore 138602, Republic of Singapore

EB, 0000-0002-8625-1139; DK, 0000-0003-3720-7462

Scaling properties of language are a useful tool for understanding generative processes in texts. We investigate the scaling relations in citywise Twitter corpora coming from the metropolitan and micropolitan statistical areas of the United States. We observe a slightly superlinear urban scaling with the city population for the total volume of the tweets and words created in a city. We then find that a certain core vocabulary follows the scaling relationship of that of the bulk text, but most words are sensitive to city size, exhibiting a super- or a sublinear urban scaling. For both regimes, we can offer a plausible explanation based on the meaning of the words. We also show that the parameters for Zipf's Law and Heaps' Law differ on Twitter from that of other texts, and that the exponent of Zipf's Law changes with city size.

## 1. Introduction

The recent increase in digitally available language corpora made it possible to extend the traditional linguistic tools to a vast amount of often user-generated texts. Understanding how these corpora differ from traditional texts is crucial in developing computational methods for web search, information retrieval or machine translation [1]. The amount of these texts enables the analysis of language on a previously unprecedented scale [2–4], including the dynamics, geography and time scale of language change [5,6], social media cursing habits [7–9] or dialectal variations [10].

From online user activity and content, it is often possible to infer different socio-economic variables on various aggregation scales. Ranging from showing correlation between the main language features on Twitter and several demographic variables [11], through predicting heart-disease rates of an area based on its language use [12] or relating unemployment to social media content and activity [13–15] to forecasting stock market moves from search semantics [16], many studies have attempted to connect online media language and metadata to real-world outcomes. Various studies have analysed spatial variation in the text of online social network messages and its applicability to several different questions, including user localization based on the content of their posts [17,18], empirical analysis of the geographical diffusion of novel words, phrases, trends and topics of interest [19,20], measuring public mood [21].

While many of the above-cited studies exploit the fact that language use or social media activity varies in space, it is hard to capture the impact of the geographical environment on the used words or concepts. There is a growing literature on how the sheer size of a settlement influences the number of patents, GDP or the total road length driven by universal laws [22]. These observations led to the establishment of the theory of urban scaling [23–31], where scaling laws with city size have been observed in various measures such as economic productivity [32], human interactions [33], urban economic diversification [34], election data [35], building heights [36], crime concentration [37,38] or touristic attractiveness [39].

In our paper, we aim to capture the effect of city size on language use via individual urban scaling laws of words. By examining the so-called scaling exponents, we are able to connect geographical size effects to systematic variations in word use frequencies. We show that the sensitivity of words to population size is also reflected in their meaning. We also investigate how social media language and city size affects the parameters of Zipf's Law [40], and how the exponent of Zipf's Law is different from that of the literature value [40,41]. We also show that the number of new words needed in longer texts (Heaps' Law [2]) exhibits a sublinear power-law form on Twitter, indicating a decelerating growth of distinct tokens with city size.

# 2. Methods

## 2.1. Twitter and census data

We use data from the online social network Twitter, which freely provides approximately 1% of all sent messages via their streaming API. For mobile devices, users have an option to share their exact location along with the Twitter message. Therefore, some messages contain geolocation information in the form of GPS coordinates. In this study, we analyse 456 million of these geolocated tweets collected between February 2012 and August 2014 from the area of the United States. We construct a geographically indexed database of these tweets, permitting the efficient analysis of regional features [42]. Using the hierarchical triangular mesh scheme for practical geographical indexing, we assigned a US county to each tweet [43,44]. County borders are obtained from the GADM database [45]. Counties are then aggregated into metropolitan and micropolitan areas using the county to metro area crosswalk file from [46]. Population data for the metropolitan statistical area (MSA) areas are obtained from [47].

There are many ways a user can post on Twitter. Because a large amount of the posts come from third-party apps such as Foursquare, we filter the messages according to their URL field. We only leave messages that have either no source URL, or their URL after the 'https://' prefix matches one of the following SQL patterns: 'twit%', 'tl.gd%' or 'path.com%'. These are most likely text messages intended for the original use of Twitter, and where automated texts such as the phrase 'I'm at' or 'check-in' on Foursquare are left out.

For the tokenization of the Twitter messages, we use the toolkit published on https://github.com/eltevo/twtoolkit. We leave out words that are less than three characters long, contain numbers or have the same consecutive character more than twice. We also filter hashtags, characters with high unicode values, usernames and web addresses [42].

## 2.2. Urban scaling

Most urban socio-economic indicators follow the certain relation for a certain urban system

$$Y(N) = Y_0 \cdot N^\beta, \qquad (2.1)$$

where $Y$ denotes a quantity (economic output, number of patents, crime rate, etc.) related to the city, $Y_0$ is a multiplication factor, $N$ is the size of the city in terms of its population and $\beta$ denotes a scaling exponent, that captures the dynamics of the change of the quantity $Y$ with city population $N$. $\beta = 1$ describes a linear relationship, where the quantity $Y$ is linearly proportional to the population, which is usually associated with individual human needs such as jobs, housing or water consumption. The case $\beta > 1$ is called superlinear scaling, and it means that larger cities exhibit disproportionately more of the quantity $Y$ than smaller cities. This type of scaling is usually related to larger cities being disproportionately the centres of innovation and wealth. The opposite case is when $\beta < 1$, that is called sublinear scaling, and is usually related to infrastructural quantities such as road network length, where urban agglomeration effects create more efficiency [27].

Here, we investigate scaling relations between urban area populations and various measures of Twitter activity and the language on Twitter. When fitting scaling relations on aggregate metrics or on the number of times a certain word appears in a metropolitan area, we always assume that the total number of tweets, or the total number of a certain word $Y_{tot}$ must be conserved in the law. That means that we have only one parameter in our fit, the value of $\beta$, while the multiplication factor $Y_0$ determined by $\beta$ and $Y_{tot}$ is as follows:

$$\sum_{i=1}^{K} Y_0 \cdot N_i^{\beta} = Y_{tot},$$

where the index $i$ denotes different cities, the total number of cities is $K$, the exponent $\beta$ is the exponent of the scaling for the investigated metrics and $N_i$ is the population of the city with index $i$.

We use the 'Person Model' of Leitão et al. [48], where this conservation is ensured by the normalization factor, and where the assumption is that out of the total number of $Y_{tot}$ units of output that exists in the whole urban system, the probability $p(j)$ for one person $j$ to obtain one unit of output depends only on the population $N_j$ of the city where person $j$ lives as

$$p(j) = \frac{N_j^{\beta-1}}{Z(\beta)},$$

where $Z(\beta)$ is the normalization constant, i.e. $Z(\beta) = \sum_{j=1}^{M} N_j^{\beta-1}$, if there are altogether $M$ people in all of the cities. Formally, this model corresponds to a scaling relationship from (2.1), where $Y_0 = Y_{tot}/Z(\beta)$. But it can also be interpreted as urban scaling being the consequence of the scaling of word choice probabilities for a single person, which has a power-law exponent of $\beta - 1$.

To assess the validity of the scaling fits of the aggregate metrics, such as for example the scaling exponent for the total number of words $\beta_{words}$, we confirm nonlinear scaling, if the difference between the likelihoods of a model with an exponent $\beta_{words} = 1$ and $\beta_{words}$ given by the fit is big enough. It means that the difference between the Bayesian information criterion (BIC) values of the two models $\Delta BIC = BIC_{\beta_{words}=1} - BIC_{\beta_{words} \neq 1}$ is sufficiently large [48]: $\Delta BIC > 6$. Otherwise, if $\Delta BIC < 0$, the linear model fits the scaling better, and between the two values, the fit is inconclusive.

In the following, we are going to denote by $\beta_w$ the scaling exponent of a given word $w$. In the case of the words, we compare the two models, where $\beta_w$ is set to the scaling exponent of the total number of words $\beta_w = \beta_{words}$, and where $\beta_w$ is calculated from the fits. If $\Delta BIC = BIC_{\beta_w=\beta_{words}} - BIC_{\beta_w \neq \beta_{words}} > 6$, then we conclude that the fit is nonlinear compared to the bulk text, and if $\Delta BIC < 0$, the $\beta_{words}$ model fits the scaling better, and between the two values, the fit is inconclusive.

## 2.3. Zipf's Law

We use the following form for Zipf's Law that is proposed in [49], and that fits the probability distribution of the word frequencies apart from the very rare words

$$p(f) = C \cdot f^{-\alpha}, \quad \text{if } f > f_{min},$$

where $\alpha$ is the exponent of the Zipf's Law, $f$ is word frequency, $f_{min}$ is the minimum word frequency above which the power-law assumption holds, $C$ is a multiplicative constant and $p(f)$ is the probability density function of the word frequencies.

We fit the probability distribution of the frequencies $p(f)$ using the `power law` package of Python [50], that uses a maximum-likelihood method based on the results of [51–53]. $f_{min}$ is the frequency for which the power-law fit is the most probable with respect to the Kolmogorov–Smirnov distance [50].

A perhaps more common form of the law connects the rank of a word and its frequency

$$f(r) = C \cdot r^{-\gamma},$$

where $r$ is the rank of a word, $C$ is a multiplicative constant, $\gamma$ is the power-law exponent and $f(r)$ is the frequency of the word at rank $r$. We use the previous form because the fitting method of [50] can only reliably tell the exponent for the tail of a distribution. In the rank-frequency case, the interesting part of the fit would be at the first few ranks, while the most common words are in the tail of the $p(f)$ distribution.

The two formulations can be easily transformed into each other (see [49]), which gives us

$$\alpha = \frac{1}{\gamma} + 1.$$

This enables us to compare our result to several others in the literature.

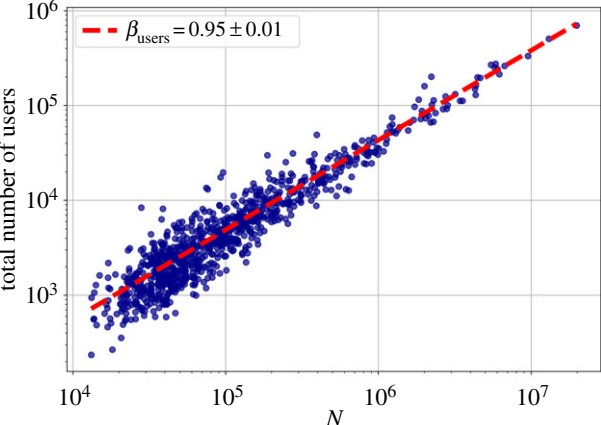

**Figure 1.** Scaling of the number of distinct users who sent a geolocated message with city population. Each point represents an MSA, the fitted line is the best maximum likelihood estimation (MLE) fit for the Person Model of [48].

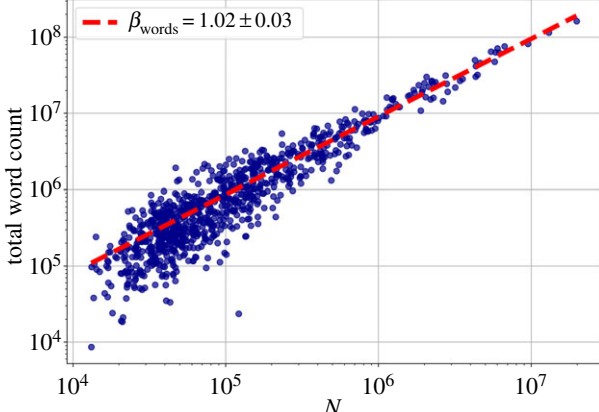

**Figure 2.** Scaling of the total number of words with city population. Each point represents an MSA, the fitted line is the best MLE fit for the Person Model of [48].

# 3. Results and discussion

## 3.1. Scaling of aggregate metrics

First, we checked how some aggregate metrics: the total number of users, the total number of individual words and the total number of tweets change with city size. Figures 1–3 show the scaling relationship data on a log–log scale, and the result of the fitted model, with exponents $\beta_{\text{users}}$, $\beta_{\text{words}}$ and $\beta_{\text{tweets}}$, respectively. In all cases, $\Delta$BIC was greater than 6, which confirmed nonlinear scaling. The total count of tweets and words both have slightly superlinear exponents $\beta_{\text{tweets}}$ and $\beta_{\text{words}}$ around 1.02. The deviation from the linear exponent may seem small, but in reality it means that for a tenfold increase in city size, the abundance of the quantity $Y$ measured increases by 5%, which is already a significant change. The number of users scales sublinearly ($\beta_{\text{users}} = 0.95 \pm 0.01$) with the city population, though. See table 1 for a summary of the fits on the aggregate metrics, together with the vocabulary size exponent from §3.4.

We have to note here, that the $\Delta$BIC values show that the nonlinear model fits the power-law scaling of the total number of words and total number of tweets significantly better than $\beta_{\text{words}} = 1$ or $\beta_{\text{users}} = 1$ would. However, the $\Delta\beta_{\text{words}}$ and the $\Delta\beta_{\text{users}}$ errors calculated from bootstrapping the original data 100 times [48] are about 0.03, that cannot exclude the linear exponent, because both $\beta_{\text{words}}$ and $\beta_{\text{users}}$ are around 1.02. The bootstrapping of the data results in a relative oversampling of the lower end of the population distribution, where deviations are higher in the data due to the large fluctuations, which might lead to an overestimated error in the scaling exponent measurements. Therefore, we conclude that $\beta_{\text{words}}$ and $\beta_{\text{tweets}}$ indicate slight superlinear scaling.

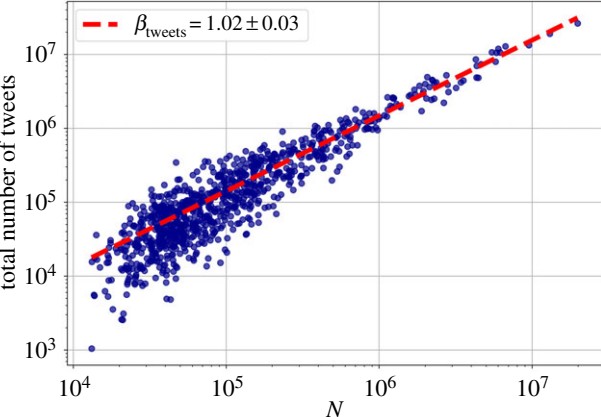

**Figure 3.** Scaling of the total number of geolocated messages with city population. Each point represents an MSA, the fitted line is the best MLE fit for the Person Model of [48].

**Table 1.** Values and bootstrapped errors of the exponent fits of the aggregated measures.

|  | $\beta$ | $\Delta\beta$ |
|---|---|---|
| total number of users ($\beta_{users}$) | 0.95 | 0.01 |
| total tweet count ($\beta_{tweets}$) | 1.02 | 0.03 |
| total tweet count ($\beta_{words}$) | 1.02 | 0.03 |
| Vocabulary size ($\beta_{vocab}$) | 0.68 | 0.01 |

It has been shown in [33] that total communication activity in human interaction networks grows superlinearly with city size. This is in line with our findings that the total number of tweets and the total word count scales superlinearly. However, the exponents are not as big as that of the number of calls or call volumes in the previously mentioned article ($\beta_{call,call\ volume} \in [1.08, 1.14]$), which suggests that scaling exponents obtained from a mobile communication network cannot automatically be translated to a social network such as Twitter.

## 3.2. Individual scaling of words

For the 11 732 words that had at least 10 000 occurrences in the dataset, we fitted scaling relationships using the Person Model with exponents $\beta_w$, where $w$ denotes an arbitrary word from the dataset. We used the exponent $\beta_{words} = 1.0207$ as an alternative model for deciding nonlinearity, because a word that has a scaling law with the same exponent as the total number of words has the same relative frequency in all urban areas. According to the $\Delta$BIC values, the fits could either be inconclusive, linear or nonlinear, with the nonlinear category being split into two by $\beta_w < 1$ sublinear, and $\beta_w > 1$ superlinear fits. The percentage of words falling into these four categories is shown in figure 4a. Most words in our Twitter corpus scale either sublinearly (45%), or superlinearly (34%) with city size. The distribution of the nonlinear exponents is visible in figure 4b. Words with a smaller exponent than $\beta_{words}$, that are 'sublinear' do not follow the text growth, thus, their relative frequency decreases as city size increases. Words with a greater exponent than $\beta_{words}$, that are 'superlinear' will relatively be more prevalent in texts in bigger cities. Three example fits from the three scaling regimes are shown in figure 5. The distribution of figure 4b also shows that even in the significantly nonlinear cases, most of the exponents are around that of the bulk text $\beta_{words}$, which means that making predictions on the expected word choice frequency of such words in individual cities based on the scaling laws might be prone to errors, and that comparing the behaviour of such words based on their exponents is not reliable near $\beta_{words}$, only in the far ends of the exponents regimes.

We sorted the words falling into the 'linear' scaling category according to their BIC values showing the goodness of fit for the fixed $\beta$ model. The first 50 words in table 2 according to this ranking are some of the most common words of the English language, apart from some swear-words and abbreviations

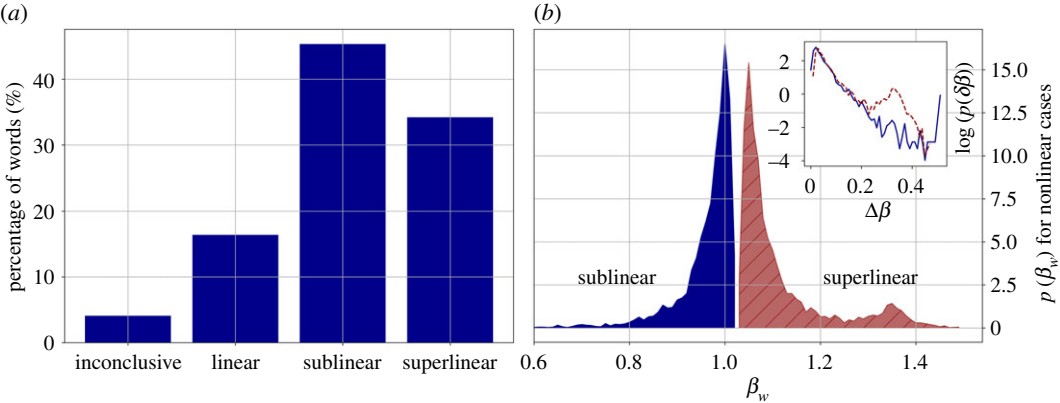

**Figure 4.** Distribution of word exponents. (*a*) Percentage of words falling into the inconclusive, linear, sublinear and superlinear scaling categories according to the ΔBIC values of fits. (*b*) Distribution of $\beta_w$ exponents for sublinear and superlinear words. The tail behaviour is in the inset, where $\delta\beta_w = |\beta_{words} - \beta_w|$.

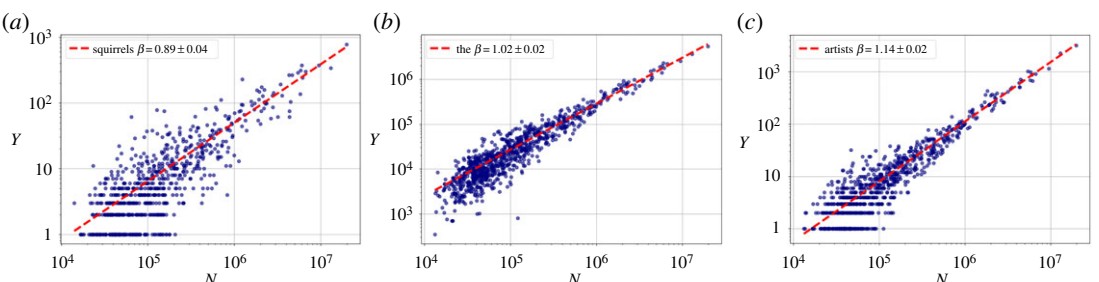

**Figure 5.** Three scaling relationships from the sublinear (*a*), linear (*b*) and superlinear (*c*) scaling regimes with the MLE fits explained in the Methods section.

**Table 2.** The top 50 words as ranked according to the BIC values for a $\beta = 1.0207$ fixed exponent Person Model. These are the words that correspond most to the scaling of the overall word volume, thus, they are the words that appear most homogeneously in the texts of all urban areas.

```
the you and that for this just lol like with have but get not your was all
   love what are when out know good now got can about one time day how they
   too shit want back need why she people right some see going today fuck
   will really her
```

(e.g. lol) that are typical for Twitter language [11]. These are the words that are most homogeneously present in the text of all urban areas.

From the first 5000 words according to word rank by occurrence, the most sublinearly and superlinearly scaling words can be seen in table 3. Their exponent differs significantly from $\beta_{words}$, and their meaning can usually be linked to the exponent range qualitatively. The sublinearly scaling words mostly correspond to weather services reporting (flood 0.54, thunderstorm 0.61, wind 0.85), some certain slang and swear-word forms (shxt 0.81, dang 0.88, damnit 0.93), outdoor-related activities (fishing 0.82, deer 0.81, truck 0.90, hunting 0.87) and certain companies (walmart 0.83). There is a longer tail with a distinctive peak in the range of superlinearly scaling words than in the sublinear regime in figure 4*b*. This tail corresponds to Spanish words (gracias 1.41, por 1.40, para 1.39, etc.), that could not be separated from the English text, since the shortness of tweets make automated language detection very noisy. Apart from the Spanish words, again some special slang or swear-words (deadass 1.52, thx 1.16, lmfao 1.17, omfg 1.16), flight-reporting (flight 1.25, delayed 1.24, etc.) and lifestyle-related words (fitness 1.15, fashion 1.15, restaurant 1.14, traffic 1.22) dominate this end of the distribution.

Thus, when compared to the slightly nonlinear scaling of total amount of words, not all words follow the growth homogeneously with this same exponent. Though a significant amount remains in the linear

**Table 3.** The most sublinearly or superlinearly scaling words out of the 5000 most frequent words with small bootstrapped error $\Delta\beta_w < 0.1$.

| word | $\beta_w$ | $\Delta\beta_w$ | word | $\beta_w$ | $\Delta\beta_w$ |
|---|---|---|---|---|---|
| advisory | 0.50 | 0.07 | hoy | 1.41 | 0.10 |
| flood | 0.54 | 0.07 | gracias | 1.41 | 0.09 |
| severe | 0.58 | 0.05 | por | 1.40 | 0.09 |
| thunderstorm | 0.61 | 0.06 | para | 1.39 | 0.10 |
| warning | 0.62 | 0.05 | feliz | 1.39 | 0.09 |
| arkansas | 0.65 | 0.10 | con | 1.39 | 0.08 |
| statement | 0.72 | 0.04 | cuando | 1.39 | 0.09 |
| April | 0.75 | 0.04 | que | 1.38 | 0.09 |
| tractor | 0.75 | 0.05 | siempre | 1.38 | 0.08 |
| February | 0.78 | 0.05 | amor | 1.37 | 0.08 |
| chapel | 0.78 | 0.09 | ver | 1.36 | 0.09 |
| bama | 0.80 | 0.09 | mejor | 1.36 | 0.08 |
| ole | 0.80 | 0.07 | bien | 1.35 | 0.09 |
| unc | 0.80 | 0.07 | jajaja | 1.35 | 0.10 |
| beside | 0.81 | 0.06 | mas | 1.35 | 0.10 |
| deer | 0.81 | 0.04 | del | 1.35 | 0.08 |
| shelby | 0.81 | 0.08 | todo | 1.35 | 0.09 |
| kentucky | 0.81 | 0.07 | tengo | 1.35 | 0.09 |
| ian | 0.82 | 0.07 | porque | 1.34 | 0.08 |
| fishing | 0.82 | 0.05 | eres | 1.34 | 0.08 |
| dorm | 0.82 | 0.04 | linda | 1.33 | 0.08 |
| freeze | 0.82 | 0.03 | muy | 1.33 | 0.09 |
| carolina | 0.83 | 0.08 | quiero | 1.33 | 0.08 |
| walmart | 0.83 | 0.05 | hola | 1.33 | 0.06 |
| December | 0.83 | 0.04 | las | 1.33 | 0.10 |
| January | 0.83 | 0.03 | mucho | 1.33 | 0.08 |
| tornado | 0.84 | 0.07 | nada | 1.33 | 0.08 |
| accounting | 0.84 | 0.06 | mañana | 1.32 | 0.09 |
| mountains | 0.85 | 0.06 | amo | 1.32 | 0.09 |
| wind | 0.85 | 0.10 | soy | 1.32 | 0.08 |
| campus | 0.85 | 0.04 | les | 1.31 | 0.07 |
| exams | 0.85 | 0.06 | hay | 1.30 | 0.09 |
| advisor | 0.85 | 0.04 | mis | 1.29 | 0.07 |
| mart | 0.85 | 0.05 | bueno | 1.28 | 0.07 |
| roommates | 0.86 | 0.05 | gusta | 1.28 | 0.07 |
| barrel | 0.86 | 0.05 | brunch | 1.28 | 0.06 |
| roads | 0.86 | 0.05 | mal | 1.27 | 0.08 |
| lmbo | 0.86 | 0.08 | museum | 1.27 | 0.07 |
| duke | 0.86 | 0.06 | uno | 1.27 | 0.08 |
| logan | 0.87 | 0.08 | flight | 1.25 | 0.07 |
| roommate | 0.87 | 0.03 | dos | 1.24 | 0.07 |

(*Continued.*)

**Table 3.** (Continued.)

| word | $\beta_w$ | $\Delta\beta_w$ | word | $\beta_w$ | $\Delta\beta_w$ |
|---|---|---|---|---|---|
| baptist | 0.87 | 0.06 | landed | 1.24 | 0.08 |
| exam | 0.87 | 0.05 | dice | 1.24 | 0.07 |
| brooke | 0.87 | 0.05 | casa | 1.24 | 0.07 |
| bahaha | 0.87 | 0.04 | grande | 1.23 | 0.06 |
| ski | 0.87 | 0.07 | fin | 1.22 | 0.06 |
| slowly | 0.87 | 0.09 | traffic | 1.22 | 0.08 |
| further | 0.87 | 0.07 | com | 1.22 | 0.05 |
| hunting | 0.87 | 0.02 | lounge | 1.21 | 0.07 |
| ymca | 0.87 | 0.04 | heights | 1.20 | 0.06 |

or inconclusive range according to the statistical model test, most words are sensitive to city size and exhibit a super- or sublinear scaling. Those that fit the linear model the best, correspond to a kind of 'core-Twitter' vocabulary, which has a lot in common with the most common words of the English language, but also shows some Twitter-specific elements. A visible group of words that are among the most super- or sublinearly scaling words are related to the abundance or lack of the elements of urban lifestyle (e.g. deer, fitness). Thus, the imprint of the physical environment appears in a quantifiable way in the growths of word occurrences as a function of urban populations. Swear-words and slang, that are quite prevalent in this type of corpus [7,8], appear at both ends of the regime that suggests that some specific forms of swearing disappear with urbanization, but the share of overall swearing on Twitter grows with city size. The peak consisting of Spanish words at the superlinear end of the exponent distribution marks the stronger presence of the biggest non-English speaking ethnicity in bigger urban areas. This is confirmed by fitting the scaling relationship to the Hispanic or Latino population of the MSA areas (the exponent fitted on the data from [54] with the methods of the paper $\beta_{\text{Hisp. population}} = 1.31 \pm 0.14$), which despite the large error, is very superlinear.

Using the census-based MSA definitions as delineations for city boundaries might have an effect on the measured exponents as well. Defining city boundaries based on different population density and commuting flow thresholds and aggregation might lead to inconsistent fits for the same urban measure such as the number of patents [25]. However, the change in the exponent $\beta$ in the cited paper is continuous in the parameter space of the aggregation, which means that our results for the different $\beta_w$ exponents would still hold when the exponents of different words are compared to each other. Moreover, the spatial distribution of Twitter activity in cities or metropolitan areas is highly concentrated on the most populous areas (see, for example, fig. 1 in [55]), which means that for this specific dataset, results would not change very much by imposing stricter boundaries or more sophisticated boundary detection algorithms, that still rely on population density [25,56].

## 3.3. Zipf's Law on Twitter

Figure 6 shows the distribution of word counts in the overall corpus. The power-law fit gave a minimum count $f_{\min} = 13$, and an exponent $\alpha = 1.682 \pm 0.001$. To check whether this law depends on city size, we fitted the same distribution for the individual cities, and according to figure 7, the exponent gradually decreases with city size, that is, it decreases with the length of the text.

That the relative frequency of some words changes with city size means that the frequency of words versus their rank, Zipf's Law, can vary from metropolitan area to metropolitan area. We obtained that the exponent of Zipf's Law depends on city size, namely that the exponent decreases as text size increases. It means that with the growth of a city, rarer words tend to appear in greater numbers. The values obtained for the Zipf exponent are in line with the theoretical bounds 1.6–2.4 of [57]. In the communication efficiency framework [57,58], decreasing $\alpha$ can be understood as decreased communication efficiency due to the increased number of different tokens, that requires more effort in the process of understanding from the reader. Using more specific words can also be a result of the 140 character limit, that was the maximum length of a tweet at the time of the data collection, and it may be a

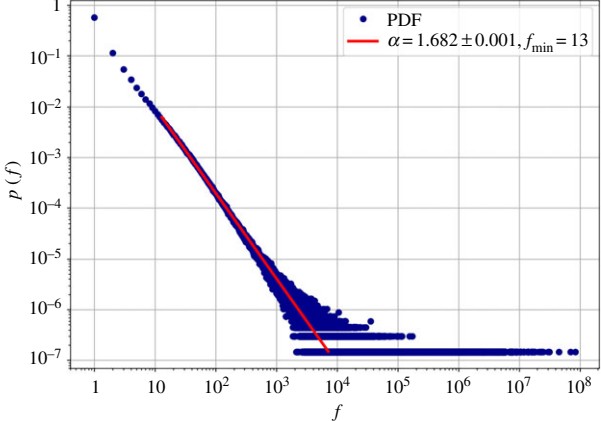

**Figure 6.** Probability distribution of word frequencies in the overall corpus and power-law fitted by the `power law` package.

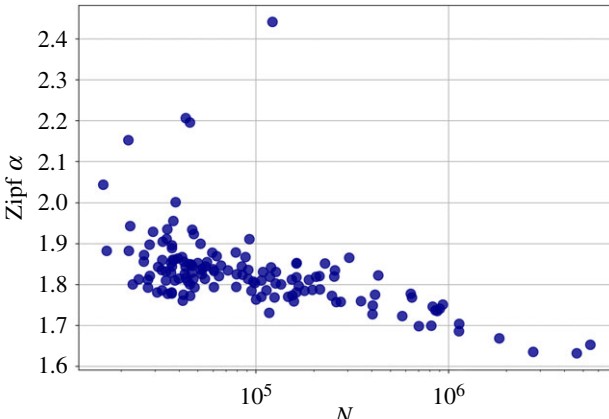

**Figure 7.** Dependency of the Zipf exponent on city population. The exponent decreases as the population, and with the population, the total number of words in a city grows.

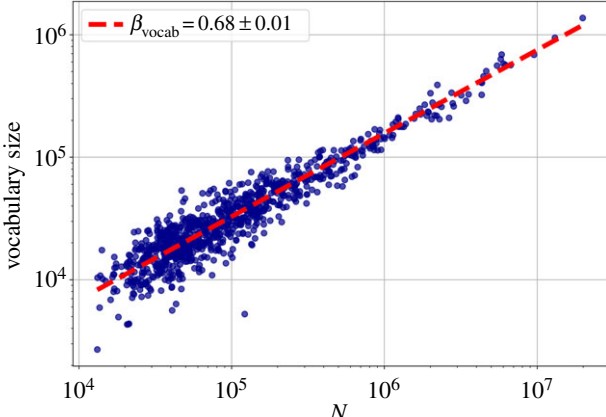

**Figure 8.** Scaling of the total number of distinct words with city population. Each point represents an MSA, the fitted line is the best MLE fit for the Person Model of [48].

similar effect to that of texting [59]. This suggests that the carrying medium has a huge impact on the exact values of the parameters of linguistic laws.

The Zipf exponent measured in the overall corpus is also much lower than the $\alpha = 2$ from the original law [40]. We do not observe the second power-law regime either, as suggested by Montemurro [60] and Ferrer i Cancho & Solé [49]. Because most observations so far hold only for books or corpora that contain

longer texts than tweets, our results suggest that the nature of communication, in our case, Twitter itself affects the parameters of linguistic laws.

## 3.4. Vocabulary size change

Figure 8 shows the vocabulary size as a function of the metropolitan area population, and the power-law fit. It shows that, contrary to the previous aggregate metrics, the vocabulary size grows sublinearly ($\beta_{\mathrm{vocab}} = 0.68$) with the city size. This relationship can also be translated to the dependency on the total word count, which would give a $\beta_{\mathrm{Heaps}} = \beta_{\mathrm{vocab}}/\beta_{\mathrm{words}} = 0.68/1.02 = 0.67$, another sublinear scaling. Sampling texts of the same length ($10^6$ words) from cities with different populations yielded almost constant vocabulary size. Thus, city size does not affect the vocabulary size considerably. Therefore, the sublinear exponent $\beta_{\mathrm{vocab}} = 0.68$, and the derived Heaps' exponent $\beta_{\mathrm{Heaps}} = 0.67$ is in line with Heaps' Law exponents found in the literature, although it might differ from the 0.49 to 0.54 range found in other English corpora [61].

## 4. Conclusion

In this paper, we investigated the scaling relations in citywise Twitter corpora coming from the metropolitan and micropolitan statistical areas of the United States. We could observe a slightly superlinear scaling decreasing with the city population for the total volume of the tweets and words created in a city. When observing the scaling of individual words, we found that a certain core vocabulary follows the scaling relationship of that of the bulk text, but most words are sensitive to city size, and their frequencies either increase at a higher or a lower rate with city size than that of the total word volume. At both ends of the spectrum, the meaning of the most superlinearly or most sublinearly scaling words is representative of their exponent. We also examined the increase in the number of distinct words with city size, which has an exponent in the sublinear range in line with Heaps' Law from linguistics.

Data accessibility. Owing to Twitter's policy, we cannot publicly share the original dataset used in this analysis. However, aggregated results from which all calculations can be recreated are available in at http://bokae.web.elte.hu/papers/2018/word_scaling and from the Dryad Digital Repository: https://doi.org/10.5061/dryad.824f24t [62].
Authors' contributions. E.B. and G.V. designed the study, E.B. and D.K. analysed the data, E.B., D.K. and G.V. synthetized the results, E.B. and D.K. wrote the manuscript. All authors gave final approval for publication and agree to be held accountable for the work performed therein.
Competing interests. The authors declare no competing interests.
Funding. The authors thank the support of the National Research, Development and Innovation Office of Hungary (grant no. KH125280).

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
