## [Reviewer comments · Royal Society Open Science]

Review History

RSOS-190027.R0 (Original submission)

Review form: Reviewer 1

Is the manuscript scientifically sound in its present form?

Yes

Are the interpretations and conclusions justified by the results?

Yes

Is the language acceptable?

Yes

Is it clear how to access all supporting data?

Yes

Do you have any ethical concerns with this paper?

No

Have you any concerns about statistical analyses in this paper?

Yes

Recommendation?

Major revision is needed (please make suggestions in comments)

Comments to the Author(s)

I have studied the manuscript "Scaling in Words on Twitter" by Bokányi, Kondor, and Vattay, submitted for publication in Royal Society Open Science.

Here the authors investigate word use on Twitter after aggregating geolocalized tweets (collected in the period 2012-2014) from the US into Metropolitan and Micropolitan Statistical Areas (CBSA - Core Based Statistical Area). For the analysis, the authors propose a parallel with the Urban Scaling Hypothesis and study scaling relationships of the number of Twitter users, total number of words, and total number of tweets all as a function of the population. The authors also investigate how the scaling law changes when considering the individual occurrence of words versus the city population. In addition to that, they also examine the Zipf's Law and how the Zipf exponent changes with the city size (CBSAs population). Finally, the authors explore the Heaps' law, finding that the vocabulary size increases sub-linearly with the city size.

I found that the work is exciting, and this idea of investigating linguistic laws in connection with the Urban Scaling Hypothesis is highly innovative.

There are, however, some points I think the authors should consider before my recommendation for publication.

- 1) There are different scaling exponents and all are named β . This makes reading confusing. Even the authors appear to acknowledge that since they use β_W on page 5, line 4, but without properly defining it. I suggest the authors clarify their notation, perhaps utilizing a subscript word for each variable (perhaps: β_{users} , β_{words} , β_{tweets} , and so on).
- 2) For the fitting, the authors rely on the MLE approach of Leitão et al. They also verify whether β is significantly different from 1 by comparing the BIC values estimated from the "Person Model" with the same model with $\beta=1$. In addition to that, I think the authors should consider the confidence intervals of β for asserting that the relationships are non-linear. For instance, the estimates of Figures 2 and 3 are both 1.02, but their standard errors are 0.03, and thus, we cannot consider those as non-linear relationships. The same problem holds for the analysis of individual words. It would also be interesting to summarize all aggregated scaling exponents into a bar plot or a table.
- 3) The authors find that β is word-dependent. That is, perhaps, the most interesting finding of this work. I suggest the authors elaborate more on that finding. For sure, they can improve the description and the presentation of the results of Section 3.2. I have some concrete suggestions I wish the authors consider.
 - i) It would be very interesting to investigate whether the value of β is dependent on the word rank. This can be done by a simple plot of β vs. word rank. I would suspect that more common words would have $\beta \approx 1$, but I don't have a clue about what differentiates among words with sub superlinear scaling laws. If something interesting comes out from this analysis, the authors may also consider a comparison with the results of Ref. [Phys. Rev. E 88, 024802 (2013)], where it was found that scaling laws of variables accounting for participation in elections depend on the importance of the political position.

ii) Figure 5 could be improved to illustrate the results of Section 3.2 better. I suggest the authors make a two-panel plot. In the first, they could show a bar plot depicting the percentage of words having linear, inconclusive, non-linear, sublinear, and superlinear scaling. In the second, they could show the probability distribution of β_{word} only for words having non-linear relationships. I would also be interesting to consider splitting the data into sublinear and superlinear, and perhaps to include an inset showing the absolute difference between 1 and β_{word} [$\text{abs}(1 - \beta_{\text{word}})$] for both cases. This would help readers to see the different tail behaviors.

4) In Figure 7, the authors use the number of words in a city as a proxy for city size. I guess it would be much better to directly use the CBSA population since the discussion is all about α changing with the city size.

5) Finally, the authors should consider acknowledging the limitations involving using CBSA as city definition. This is an important issue widely discussed in the Science of City community, and the authors are probably aware of that since they cite Refs. [25], [26] and [48]. I guess a single paragraph pointing out this limitation of the present work is enough, but eventually, the authors may consider using a clustering approach for defining urban units in future studies (for instance, the CCA - city clustering algorithm).

Minor points:

- page 2, line 17: Please, define OSN;
- page 2, lines 47-48: the statement "exhibits a power-law form on Twitter, indicating a decelerating growth of distinct tokens with city size" appears incomplete. I guess the authors wish to mention about the sublinear relationship. Please, clarify;
- page 4, lines 5-6: Please, fix the double use of "and";
- page 4, line 19: Ref. [27] is cited after the period;
- page 5, line 44: Please, close the bracket after Ref. [49];
- page 5, Equations: Please, define each variable as you did for Eq. (1).
- page 8, line 15: Figure 5 is cited before Figure 4;
- page 9, Figure 5: the y-axis label is truncated;
- page 9-10, Tables 1 and 2: Please, make a standard table with fewer words. The way these tables are presented is weird and confusing. These new tables could show each value of β .
- page 11, lines 42-43: There is a mention to an SI. Since this paper has no SI, I'm assuming the mention is about Ref. [54], but that is a web page. Please, clarify;
- page 12, line 40: In the statement "decreasing β can be understood", it should be α .

Review form: Reviewer 2 (Eduardo G. Altmann)

Is the manuscript scientifically sound in its present form?

Yes

Are the interpretations and conclusions justified by the results?

Yes

Is the language acceptable?

Yes

Is it clear how to access all supporting data?

Yes

Do you have any ethical concerns with this paper?

No

Have you any concerns about statistical analyses in this paper?

No

Recommendation?

Accept with minor revision (please list in comments)

Comments to the Author(s)

The manuscript "Scaling in Words on Twitter" investigates how the activity and language of Twitter varies with the population size of the Urban area in which it was published. This investigation involves the interplay between different scaling laws that have been extensively studied in recent years (in the field of complex systems): urban scaling laws and scaling laws in natural language. The manuscript uses state of the art statistical methods (Maximum Likelihood inference) and reports very interesting findings: how the frequency of specific words scales with the population of different US cities. The distinction between the most super- and sub-linear words done in Tables 1 and 2 is remarkable. The scalings of these words reflect both scaling properties of language more generally and scaling properties of the objects being referred by the words, a fascinating example of interplay between different complex systems. In summary, I think the methods employed in this manuscript are sound and the findings are new and well justified. I recommend publication after the authors consider the points below.

1) The authors emphasize that most of the observed exponents β are statistically different from $\beta=1$. While I believe the statistical analysis supporting this conclusion is correct, this does not imply anything about the strength of the non-linearity. Given the large fluctuation that exists around the average behaviour, exponents β close to $\beta=1$ end up having a very little influence on what can be said about individual cities.

2) In Secs. 3.3 and 3.4, I found that the distinction between the role of database size and of city size was not clear. The volume of twitter activity (and thus the number of word tokens) scales roughly linear with city size (as shown in Fig. 2). Therefore, the result in Fig. 8 is essentially due to Heaps' law and not due to city size. The authors finish Sec. 3.4 saying that "the decelerated extension of the vocabulary in bigger cities...". I think this sentence (and some of the previous presentation, also at the end of Sec. 3.3) is misleading because this effect is not due to population size. It is due to the size of your dataset in each city (number of tweets, or word tokens). This size is ultimately related to the amount of data you collected, which is an arbitrary choice. In order to claim the effect of city size, one would need to control for database size, e.g., choosing the same

number of word tokens from cities of different sizes and then counting in which ones there are more word types.

Minor points and typos:

-page 5, line 44 a bracket ")" is missing after [49]

-p. 6, line 15 "The the" -> "The"

-p. 11, line 53. Instead of "x_min" one should write "f_min"? Also in the caption of Fig. 6.

- p. 12, lines 40 and 50. Instead of "\beta" you mean "\alpha" (the Zipfian exponent)?

Decision letter (RSOS-190027.R0)

09-Jul-2019

Dear Ms Bokányi,

The editors assigned to your paper ("Scaling in Words on Twitter") have now received comments from reviewers. We would like you to revise your paper in accordance with the referee and Associate Editor suggestions which can be found below (not including confidential reports to the Editor). Please note this decision does not guarantee eventual acceptance.

Please submit a copy of your revised paper before 01-Aug-2019. Please note that the revision deadline will expire at 00.00am on this date. If we do not hear from you within this time then it will be assumed that the paper has been withdrawn. In exceptional circumstances, extensions may be possible if agreed with the Editorial Office in advance. We do not allow multiple rounds of revision so we urge you to make every effort to fully address all of the comments at this stage. If deemed necessary by the Editors, your manuscript will be sent back to one or more of the original reviewers for assessment. If the original reviewers are not available, we may invite new reviewers.

- Data accessibility

If you wish to submit your supporting data or code to Dryad (<http://datadryad.org/>), or modify your current submission to dryad, please use the following link:
<http://datadryad.org/submit?journalID=RSOS&manu=RSOS-190027>

- Competing interests

- Authors' contributions

- Acknowledgements

- Funding statement

on behalf of Dr Hamed Haddadi (Associate Editor) and Marta Kwiatkowska (Subject Editor)
 openscience@royalsociety.org

Associate Editor's comments (Dr Hamed Haddadi):

Thanks for submitting your interesting work to RSOS. Please visit the reviewers' recommendations and update the manuscript accordingly to prepare an improved submission.

Comments to Author:

Reviewers' Comments to Author:

Reviewer: 1

Comments to the Author(s)

I have studied the manuscript "Scaling in Words on Twitter" by Bokányi, Kondor, and Vattay, submitted for publication in Royal Society Open Science.

Here the authors investigate word use on Twitter after aggregating geolocalized tweets (collected in the period 2012-2014) from the US into Metropolitan and Micropolitan Statistical Areas (CBSA - Core Based Statistical Area). For the analysis, the authors propose a parallel with the Urban Scaling Hypothesis and study scaling relationships of the number of Twitter users, total number of words, and total number of tweets all as a function of the population. The authors also investigate how the scaling law changes when considering the individual occurrence of words versus the city population. In addition to that, they also examine the Zipf's Law and how the Zipf exponent changes with the city size (CBSAs population). Finally, the authors explore the Heaps' law, finding that the vocabulary size increases sub-linearly with the city size.

I found that the work is exciting, and this idea of investigating linguistic laws in connection with the Urban Scaling Hypothesis is highly innovative.

There are, however, some points I think the authors should consider before my recommendation for publication.

1) There are different scaling exponents and all are named β . This makes reading confusing. Even the authors appear to acknowledge that since they use β_W on page 5, line 4, but without properly defining it. I suggest the authors clarify their notation, perhaps utilizing a subscript word for each variable (perhaps: β_{users} , β_{words} , β_{tweets} , and so on).

2) For the fitting, the authors rely on the MLE approach of Leitão et al. They also verify whether β is significantly different from 1 by comparing the BIC values estimated from the "Person Model" with the same model with $\beta=1$. In addition to that, I think the authors should consider the confidence intervals of β for asserting that the relationships are non-linear. For instance, the estimates of Figures 2 and 3 are both 1.02, but their standard errors are 0.03, and thus, we cannot consider those as non-linear relationships. The same problem holds for the analysis of individual words. It would also be interesting to summarize all aggregated scaling exponents into a bar plot or a table.

3) The authors find that β is word-dependent. That is, perhaps, the most interesting finding of this work. I suggest the authors elaborate more on that finding. For sure, they can improve the

description and the presentation of the results of Section 3.2. I have some concrete suggestions I wish the authors consider.

i) It would be very interesting to investigate whether the value of β is dependent on the word rank. This can be done by a simple plot of β vs. word rank. I would suspect that more common words would have $\beta \approx 1$, but I don't have a clue about what differentiates among words with sub superlinear scaling laws. If something interesting comes out from this analysis, the authors may also consider a comparison with the results of Ref. [Phys. Rev. E 88, 024802 (2013)], where it was found that scaling laws of variables accounting for participation in elections depend on the importance of the political position.

ii) Figure 5 could be improved to illustrate the results of Section 3.2 better. I suggest the authors make a two-panel plot. In the first, they could show a bar plot depicting the percentage of words having linear, inconclusive, non-linear, sublinear, and superlinear scaling. In the second, they could show the probability distribution of β_{word} only for words having non-linear relationships. I would also be interesting to consider splitting the data into sublinear and superlinear, and perhaps to include an inset showing the absolute difference between 1 and β_{word} [$\text{abs}(1 - \beta_{\text{word}})$] for both cases. This would help readers to see the different tail behaviors.

4) In Figure 7, the authors use the number of words in a city as a proxy for city size. I guess it would be much better to directly use the CBSA population since the discussion is all about α changing with the city size.

5) Finally, the authors should consider acknowledging the limitations involving using CBSA as city definition. This is an important issue widely discussed in the Science of City community, and the authors are probably aware of that since they cite Refs. [25], [26] and [48]. I guess a single paragraph pointing out this limitation of the present work is enough, but eventually, the authors may consider using a clustering approach for defining urban units in future studies (for instance, the CCA - city clustering algorithm).

Minor points:

- page 2, line 17: Please, define OSN;

- page 2, lines 47-48: the statement "exhibits a power-law form on Twitter, indicating a decelerating growth of distinct tokens with city size" appears incomplete. I guess the authors wish to mention about the sublinear relationship. Please, clarify;

- page 4, lines 5-6: Please, fix the double use of "and";

- page 4, line 19: Ref. [27] is cited after the period;

- page 5, line 44: Please, close the bracket after Ref. [49];

- page 5, Equations: Please, define each variable as you did for Eq. (1).

- page 8, line 15: Figure 5 is cited before Figure 4;

- page 9, Figure 5: the y-axis label is truncated;

- page 9-10, Tables 1 and 2: Please, make a standard table with fewer words. The way these tables are presented is weird and confusing. These new tables could show each value of β .

- page 11, lines 42-43: There is a mention to an SI. Since this paper has no SI, I'm assuming the mention is about Ref. [54], but that is a web page. Please, clarify;

- page 12, line 40: In the statement "decreasing β can be understood", it should be α .

Reviewer: 2

Comments to the Author(s)

The manuscript "Scaling in Words on Twitter" investigates how the activity and language of Twitter varies with the population size of the Urban area in which it was published. This investigation involves the interplay between different scaling laws that have been extensively studied in recent years (in the field of complex systems): urban scaling laws and scaling laws in natural language. The manuscript uses state of the art statistical methods (Maximum Likelihood inference) and reports very interesting findings: how the frequency of specific words scales with the population of different US cities. The distinction between the most super- and sub-linear words done in Tables 1 and 2 is remarkable. The scalings of these words reflect both scaling properties of language more generally and scaling properties of the objects being referred by the words, a fascinating example of interplay between different complex systems. In summary, I think the methods employed in this manuscript are sound and the findings are new and well justified. I recommend publication after the authors consider the points below.

1) The authors emphasize that most of the observed exponents β are statistically different from $\beta=1$. While I believe the statistical analysis supporting this conclusion is correct, this does not imply anything about the strength of the non-linearity. Given the large fluctuation that exists around the average behaviour, exponents β close to $\beta=1$ end up having a very little influence on what can be said about individual cities.

2) In Secs. 3.3 and 3.4, I found that the distinction between the role of database size and of city size was not clear. The volume of twitter activity (and thus the number of word tokens) scales roughly linear with city size (as shown in Fig. 2). Therefore, the result in Fig. 8 is essentially due to Heaps' law and not due to city size. The authors finish Sec. 3.4 saying that "the decelerated extension of the vocabulary in bigger cities...". I think this sentence (and some of the previous presentation, also at the end of Sec. 3.3) is misleading because this effect is not due to population size. It is due to the size of your dataset in each city (number of tweets, or word tokens). This size is ultimately related to the amount of data you collected, which is an arbitrary choice. In order to claim the effect of city size, one would need to control for database size, e.g., choosing the same number of word tokens from cities of different sizes and then counting in which ones there are more word types.

Minor points and typos:

-page 5, line 44 a bracket ")" is missing after [49]

-p. 6, line 15 "The the" -> "The"

-p. 11, line 53. Instead of " x_{\min} " one should write " f_{\min} "? Also in the caption of Fig. 6.

- p. 12, lines 40 and 50. Instead of " β " you mean " α " (the Zipfian exponent)?

Author's Response to Decision Letter for (RSOS-190027.R0)

See Appendix A.

RSOS-190027.R1 (Revision)

Review form: Reviewer 1

Is the manuscript scientifically sound in its present form?

Yes

Are the interpretations and conclusions justified by the results?

Yes

Is the language acceptable?

Yes

Do you have any ethical concerns with this paper?

No

Have you any concerns about statistical analyses in this paper?

Yes

Recommendation?

Accept as is

Comments to the Author(s)

All my comments were adequately addressed. I'm happy to reiterate my positive impression of this work and recommend its publication in present form.

Please, notice that references are not correctly appearing in the current version.

Review form: Reviewer 2 (Eduardo G. Altmann)

Is the manuscript scientifically sound in its present form?

Yes

Are the interpretations and conclusions justified by the results?

Yes

Is the language acceptable?

Yes

Do you have any ethical concerns with this paper?

No

Have you any concerns about statistical analyses in this paper?

No

Recommendation?

Accept as is

Comments to the Author(s)

The authors addressed all my concerns, I recommend publication.

Decision letter (RSOS-190027.R1)

08-Sep-2019

Dear Ms Bokányi,

I am pleased to inform you that your manuscript entitled "Scaling in Words on Twitter" is now accepted for publication in Royal Society Open Science.

Kind regards,

Andrew Dunn

on behalf of Prof Marta Kwiatkowska (Subject Editor)

Reviewer comments to Author:

Reviewer: 2

Comments to the Author(s)

The authors addressed all my concerns, I recommend publication.

Reviewer: 1

Comments to the Author(s)

All my comments were adequately addressed. I'm happy to reiterate my positive impression of this work and recommend its publication in present form.

Please, notice that references are not correctly appearing in the current version.

Appendix A

Dear Editor,

please find our detailed point-by-point answers to the reviewers' comments below.

Yours sincerely,

Eszter Bokanyi

Reviewer: 1

Comments to the Author(s)

I have studied the manuscript "Scaling in Words on Twitter" by Bokányi, Kondor, and Vattay, submitted for publication in Royal Society Open Science.

Here the authors investigate word use on Twitter after aggregating geolocalized tweets (collected in the period 2012-2014) from the US into Metropolitan and Micropolitan Statistical Areas (CBSA - Core Based Statistical Area). For the analysis, the authors propose a parallel with the Urban Scaling Hypothesis and study scaling relationships of the number of Twitter users, total number of words, and total number of tweets all as a function of the population. The authors also investigate how the scaling law changes when considering the individual occurrence of words versus the city population. In addition to that, they also examine the Zipf's Law and how the Zipf exponent changes with the city size (CBSAs population). Finally, the authors explore the Heaps' law, finding that the vocabulary size increases sub-linearly with the city size.

I found that the work is exciting, and this idea of investigating linguistic laws in connection with the Urban Scaling Hypothesis is highly innovative.

There are, however, some points I think the authors should consider before my recommendation for publication.

1) There are different scaling exponents and all are named β . This makes reading confusing. Even the authors appear to acknowledge that since they use β_W on page 5, line 4, but without properly defining it. I suggest the authors clarify their notation, perhaps utilizing a subscript word for each variable (perhaps: β_{users} , β_{words} , β_{tweets} , and so on).

We've revised the notation, and added subscripts to the exponents denoting the different scaling laws, e.g. β_{word} , β_{user} , and β_w stands for the exponent of the individual word w . These subscripts were also added to the figure legends.

2) For the fitting, the authors rely on the MLE approach of Leitão et al. They also verify whether β is significantly different from 1 by comparing the BIC values estimated from the "Person Model" with the same model with $\beta=1$. In addition to that, I think the authors should consider the confidence intervals of β for asserting that the relationships are non-linear. *For instance, the estimates of Figures 2 and 3 are both 1.02, but their standard errors are 0.03, and thus, we cannot consider those as non-linear relationships.* The same problem holds for the analysis of individual words. It would also be interesting to summarize all aggregated scaling exponents into a bar plot or a table.

Even when there is only a small deviation from $\beta=1$ in the estimated exponents, the Δ BIC criterion allows for significance testing. This way, small effects might also be labelled as having a significant deviation from the linear exponent. The error range given in the paper is calculated from bootstrapping the data 100 times, which oversamples the lower end of the city population distribution due to the heterogeneous city size distribution. This lower end contains much more fluctuation than bigger cities, which might lead to an increased error term. We admit that it is not possible to totally exclude that the measured exponent might be linear, therefore, we added a paragraph discussing the significance and the error estimation issue to the manuscript.

We've added a table summarizing the aggregated exponents and their errors.

3) The authors find that β is word-dependent. That is, perhaps, the most interesting finding of this work. I suggest the authors elaborate more on that finding. For sure, they can improve the description and the presentation of the results of Section 3.2. I have some concrete suggestions I wish the authors consider.

We updated the section following the reviewer recommendations, it has been slightly rewritten. The changes can be followed in the highlighted version of the manuscript.

i) It would be very interesting to investigate whether the value of β is dependent on the word rank. This can be done by a simple plot of β vs. word rank. I would suspect that more common words would have $\beta \approx 1$, but I don't have a clue about what differentiates among words with sub superlinear scaling laws. If something interesting comes out from this analysis, the authors may also consider a comparison with the results of Ref. [Phys. Rev. E 88, 024802 (2013)], where it was found that scaling laws of variables accounting for participation in elections depend on the importance of the political position.

We already considered the possibility, and the suggested plot is inserted below. The blue dots are the individual β - rank pairs, the black line and the shadowed area represent a sliding window average and standard deviation for 50 neighboring ranks. While it is true that for the first few ranks, that is, for the most common words, the exponent is around 1.0207, which is the same as β_{words} in the paper, after that, there is only noise in the black line, and no clear trend is distinguishable.

ii) Figure 5 could be improved to illustrate the results of Section 3.2 better. I suggest the authors make a two-panel plot. In the first, they could show a bar plot depicting the percentage of words having linear, inconclusive, non-linear, sublinear, and superlinear scaling. In the second, they could show the probability distribution of β_{word} only for words having non-linear relationships. I would also be interesting to consider splitting the data into sublinear and superlinear, and perhaps to include an inset showing the absolute difference between 1 and β_{word} [$\text{abs}(1 - \beta_{\text{word}})$] for both cases. This would help readers to see the different tail behaviors.

We've reconstructed the figure considering the suggestions, and rewrote the caption and some of the surrounding manuscript text to match the figure. This is the new two-panel figure now, indicating the breakdown into percentages of words according to the goodness of fit categories (A), the distribution of the sublinear and superlinear exponents (B), and the tail behaviour using a logarithmic scale for better readability (C). $\Delta \beta$ is the deviation from the bulk text exponent, which is $|\beta_w - 1.0207|$.

4) In Figure 7, the authors use the number of words in a city as a proxy for city size. I guess it would be much better to directly use the CBSA population since the discussion is all about α changing with the city size.

We've changed the figure with CBSA population being on the horizontal axis.

5) Finally, the authors should consider acknowledging the limitations involving using CBSA as city definition. This is an important issue widely discussed in the Science of City community, and the authors are probably aware of that since they cite Refs. [25], [26] and [48]. I guess a single paragraph pointing out this limitation of the present work is enough, but eventually, the authors may consider using a clustering approach for defining urban units in future studies (for instance, the CCA - city clustering algorithm).

We've added a paragraph discussing the possible influence of metro area delineations on the measured exponents, see the highlighted corrected manuscript.

Minor points:

We've corrected all the minor points mentioned below. We changed the order of Figure 4 and Figure 5, because of the citation order, and we exchanged the former Figure 5 with a new version based on the suggestions of Reviewer 2. We also included a detailed table instead of Table 2 containing the exponent values and their bootstrapped errors for the far ends of the exponent distribution.

- page 2, line 17: Please, define OSN;
- page 2, lines 47-48: the statement "exhibits a power-law form on Twitter, indicating a decelerating growth of distinct tokens with city size" appears incomplete. I guess the authors wish to mention about the sublinear relationship. Please, clarify;
- page 4, lines 5-6: Please, fix the double use of "and";
- page 4, line 19: Ref. [27] is cited after the period;
- page 5, line 44: Please, close the bracket after Ref. [49];

- page 5, Equations: Please, define each variable as you did for Eq. (1).
- page 8, line 15: Figure 5 is cited before Figure 4;
- page 9, Figure 5: the y-axis label is truncated;
- page 9-10, Tables 1 and 2: Please, make a standard table with fewer words. The way these tables are presented is weird and confusing. These new tables could show each value of β .
- page 11, lines 42-43: There is a mention to an SI. Since this paper has no SI, I'm assuming the mention is about Ref. [54], but that is a web page. Please, clarify;
- page 12, line 40: In the statement "decreasing β can be understood", it should be α .

Reviewer: 2

Comments to the Author(s)

The manuscript "Scaling in Words on Twitter" investigates how the activity and language of Twitter varies with the population size of the Urban area in which it was published. This investigation involves the interplay between different scaling laws that have been extensively studied in recent years (in the field of complex systems): urban scaling laws and scaling laws in natural language. The manuscript uses state of the art statistical methods (Maximum Likelihood inference) and reports very interesting findings: how the frequency of specific words scales with the population of different US cities. The distinction between the most super- and sub-linear words done in Tables 1 and 2 is remarkable. The scalings of these words reflect both scaling properties of language more generally and scaling properties of the objects being referred by the words, a fascinating example of interplay between different complex systems. In summary, I think the methods employed in this manuscript are sound and the findings are new and well justified. I recommend publication after the authors consider the points below.

1) The authors emphasize that most of the observed exponents β are statistically different from $\beta=1$. While I believe the statistical analysis supporting this conclusion is correct, *this does not imply anything about the strength of the non-linearity*. Given the large fluctuation that exists around the average behaviour, exponents β close to $\beta=1$ end up having a very little influence on what can be said about individual cities.

We've added a paragraph discussing the significance and the measurement error of the exponents. We've also extended the discussion with some reflection on most of the words being near the linear scaling range. We believe, that this does not modify our results, namely that a significant share of the Twitter vocabulary users post is sensitive to the urbanization level of people's environment, that we capture through city size.

2) In Secs. 3.3 and 3.4, I found that the distinction between the role of database size and of city size was not clear. The volume of twitter activity (and thus the number of word tokens)

scales roughly linear with city size (as shown in Fig. 2). *Therefore, the result in Fig. 8 is essentially due to Heaps' law and not due to city size.* The authors finish Sec. 3.4 saying that "the decelerated extension of the vocabulary in bigger cities...". I think this sentence (and some of the previous presentation, also at the end of Sec. 3.3) is misleading because this effect is not due to population size. It is due to the size of your dataset in each city (number of tweets, or word tokens). This size is ultimately related to the amount of data you collected, which is an arbitrary choice. *In order to claim the effect of city size, one would need to control for database size, e.g., choosing the same number of word tokens from cities of different sizes and then counting in which ones there are more word types.*

We've reconsidered this section according to the suggestions, and checked how city size affects the vocabulary size of sampled texts of the same length. Because city size has a negligible effect on vocabulary growth, we've removed and added some sentences in the paper that underline that the observed sublinear exponent β_{vocab} is attributable to the standard Heaps law for text corpora. The value of the Heaps exponent differs somewhat from literature values, though, as presented in the results section. Here's the figure for the 323 metropolitan areas with more than 10^6 words, we sampled a corpus of length 10^6 from each of the cities 10 times, and averaged their vocabulary sizes. The fitted exponent of this relationship (note the log-lin scales on the x and y axes) is 0.05, which means that while the city size increases two magnitudes, the vocabulary size remains in the same order range.

Minor points and typos:

Thank you very much for pointing them out, we've corrected all of these points.

- page 5, line 44 a bracket ")" is missing after [49]
- p. 6, line 15 "The the" -> "The"
- p. 11, line 53. Instead of "x_min" one should write "f_min"? Also in the caption of Fig. 6.
- p. 12, lines 40 and 50. Instead of "\beta" you mean "\alpha" (the Zipfian exponent)?